# Experimental observation of topological $Z_2$ exciton-polaritons in transition metal dichalcogenide monolayers

Mengyao Li[1,2,3], Ivan Sinev [4], Fedor Benimetskiy[4], Tatyana Ivanova[4], Ekaterina Khestanova[4], Svetlana Kiriushechkina[1], Anton Vakulenko[1], Sriram Guddala[1,2], Maurice Skolnick[4,5], Vinod M. Menon [2,3], Dmitry Krizhanovskii [4,5], Andrea Alù [1,3,6], Anton Samusev [4] & Alexander B. Khanikaev [1,2,3✉]

The rise of quantum science and technologies motivates photonics research to seek new platforms with strong light-matter interactions to facilitate quantum behaviors at moderate light intensities. Topological polaritons (TPs) offer an ideal platform in this context, with unique properties stemming from resilient topological states of light strongly coupled with matter. Here we explore polaritonic metasurfaces based on 2D transition metal dichalcogenides (TMDs) as a promising platform for topological polaritonics. We show that the strong coupling between topological photonic modes of the metasurface and excitons in TMDs yields a topological polaritonic $Z_2$ phase. We experimentally confirm the emergence of one-way spin-polarized edge TPs in metasurfaces integrating $MoSe_2$ and $WSe_2$. Combined with the valley polarization in TMD monolayers, the proposed system enables an approach to engage the photonic angular momentum and valley and spin of excitons, offering a promising platform for photonic/solid-state interfaces for valleytronics and spintronics.

[1] Department of Electrical Engineering, City College of New York, New York, NY, USA. [2] Physics Department, City College of New York, New York, NY, USA. [3] Physics Program, Graduate Center of the City University of New York, New York, NY, USA. [4] Department of Physics and Engineering, ITMO University, Saint Petersburg, Russia. [5] Department of Physics and Astronomy, University of Sheffield, Sheffield, UK. [6] Photonics Initiative, Advanced Science Research Center, City University of New York, New York, NY, USA. ✉email: akhanikaev@ccny.cuny.edu

Topological photonics[1–3] has seen a tremendous progress in the past years with numerous topological phases implemented in a variety of platforms, from microwave to optical frequencies[4–15]. Enriching topological photonics by mixing light with condensed matter provides even more exciting avenues for controlling exotic states of light and matter. Indeed, integrating topological photonic systems with quantum wells and quantum dots has already led to major breakthroughs, such as topological lasers[16–19], topological polaritonic phases[20–24], active[25,26] and nonlinear[27–29] topological photonic devices. Consistent with their non-topological cousins, TPs[20,21,30,31] represent "half-light and half-matter" excitations emerging as the result of strong coupling[32–36] between electromagnetic and solid-state degrees of freedom. In addition, they are enriched by topological features. The combination of photonic topological properties (one-way pseudo-spin-polarized transport, topological protection against scattering) and strong interactions arising from light-matter hybridization, may support phenomena such as topological solitons, modulation instability, and generation of squeezed topological light[12,27,29,30,37–44]. Moreover, TPs pave the way towards the development of active topological nanophotonic devices with giant optical nonlinearity[45,46] enabling control of light by light at small intensities, down to the single-photon level[47–50]. Overall, polaritonic systems[51–53] serve as an ideal interface between photonics and solid-state systems, facilitating control of spin- and valley-degrees of freedom[54–62] in future quantum devices. TPs, enriched with additional degrees of freedom, inherited from nanoscale structured photonic materials thus offer uniquely versatile control of quantum states with photons.

In this context, TPs have been an active subject of research with several recent theoretical and experimental findings, in particular in systems based on quantum wells integrated into photonic nanostructures[16,20,21,23,24,26]. However, the topological polaritonic systems reported so far have been mostly limited to 1D systems, and a 2D system characterized by 1D topological invariants[16,24]. The only TP system characterized by a 2D topological invariant[21], the Chern number, has been demonstrated recently by Klembt et al.[22] for the case of broken time-reversal (TR) symmetry. Specifically, 2D TPs have been realized in Bragg micropillar lattices based on GaAs quantum wells under the application of a strong external magnetic field. On the other hand, spin-Hall TPs, a different 2D topological polaritonic phase that does not require magnetic field and therefore has significantly more opportunities for broad applicability and integration in nanophotonic systems, has so far evaded experimental realization.

In this work we put forward an approach to spin-Hall topological polaritonics based on the versatile platform of polaritonic metasurfaces containing monolayer transition metal dichalcogenides (TMDs). Our approach leverages the large exciton dipole moment in a monolayer semiconductor and the remarkable compatibility of 2D materials with various photonic structures to realize strong coupling between light and matter. We show that the strong coupling regime between a topological spin-Hall photonic metasurface and a TMD monolayer featuring a pair of degenerate TR partner excitons gives rise to a topological transition and the formation of a topological polaritonic phase characterized by nonvanishing spin-Chern numbers. Introduction of domain walls separating topological and trivial phases is then shown to produce spin-polarized polaritonic boundary modes. Spin-locking of these modes and their selective coupling to circularly polarized light of opposite handedness enables unique polaritonic spin-Hall phenomena that we demonstrate experimentally. In addition, by studying photoluminescence of WSe$_2$ topological metasurface, we confirm valley polarization of edge polaritons.

## Results

**Topological polaritonic metasurface.** The structure we consider here is schematically shown in Fig. 1a and represents a Si photonic metasurface[12,63] supporting photonic topological spin-Hall-like phase[64] with MoSe$_2$ and WSe$_2$ TMD monolayer placed on top of it. The leaky character of the modes allows a direct excitation and probing both photonic and polaritonic modes supported by such a metasurface[65,66]. A TMD monolayer encapsulated with a thin hBN layer is placed on top of the metasurface which was adjusted to hold topological modes near the exciton frequencies. Apart from the general purpose of enhancing the quality of the exciton in the monolayer, the hBN layer enables tuning of the parameters of our system (see Supplementary Note 7).

In order to induce a topological transition in the structure[63,64,67], its symmetry is reduced by shrinking (Fig. 1a, blue unit cells) or expanding the six nearest triangular holes (Fig. 1a, red unit cells), which leads to the opening of trivial and topological photonic band gaps, respectively[12,66–68]. It is important to mention that the topological polaritonic phase reported here is enabled by the structure of the modes of the metasurface. Thus, the pairs of upper and lower cones in unperturbed structure correspond to the clockwise and counterclockwise $s = \pm 1$ circularly polarized dipolar $p_\pm = p_x \pm i p_y (l = \pm 1 = 1 \times s)$ and quadrupolar $d_\pm = d_{xy} \pm i d_{x^2-y^2}$ ($l = \pm 2 = 2 \times s$) modes, where the polarization handedness $s$ takes the role of a photonic pseudo-spin, essential for the spin-Hall phase[64].

By adding a TMD monolayer on top of this photonic structure we introduce excitonic degrees of freedom[67,69]. It is important to stress that excitons in TMDs (MoSe$_2$ and WSe$_2$ in our case) are (i) characterized by non-zero angular momentum (the spin of s-excitons) of $m = +1$ and $m = -1$ (at K and K' valleys, respectively), and thus form TR partners essential for the topological polaritonic spin-Hall phase engineered here, and (ii) polarized in the plane of the monolayer, which allows to efficiently couple the in-plane electric field of the modes of the metasurfaces to the excitons[65]. We note that the characteristic size of excitons (~1 nm) and the scale of spatial variation of the photonic modes are orders of magnitude apart, which implies that excitons can interact with both dipolar and quadrupolar photonic modes, and there are no selection rules with respect to the orbital momentum of photonic modes. Nonetheless, the selection rules with respect to the photon pseudo-spin $s$ do apply to interactions of optical and excitonic modes due to the conservation of angular momentum (see discussion in Supplementary Note 4).

**Theoretical description.** The topological polaritonic system under study can be described by an effective Hamiltonian of the form

$$\widehat{\mathcal{H}} = \widehat{\mathcal{H}}_{ph} + \widehat{\mathcal{H}}_{ex} + \sum_{\mathbf{k},l,m,s} (g_{(\mathbf{k},l,m,s)} \hat{b}_m^+ \hat{a}_{\mathbf{k},l,s} \delta_{s,m} + g^+_{(\mathbf{k},l,m,s)} \hat{b}_m \hat{a}^+_{\mathbf{k},l,s} \delta_{s,m}), \quad (1)$$

$$\widehat{\mathcal{H}}_{ph} = \sum_{\mathbf{k},l,l',s,s'} \widehat{H}_{(\mathbf{k},l,l',s,s')} \hat{a}^+_{\mathbf{k},l',s'} \hat{a}_{\mathbf{k},l,s}, \quad (2)$$

$$\widehat{\mathcal{H}}_{ex} = \sum_{m,m'} \tilde{\omega}_{ex} \hat{b}^+_{m'} \hat{b}_m, \quad (3)$$

where the photonic Hamiltonian $\widehat{\mathcal{H}}_{ph}$ is obtained from electromagnetic theory (detailed in Supplementary Note 2) and it assumes the form of the well-known Bernevig–Hughes–Zhang (BHZ) Hamiltonian[70] of the two-dimensional $Z_2$ topological insulator

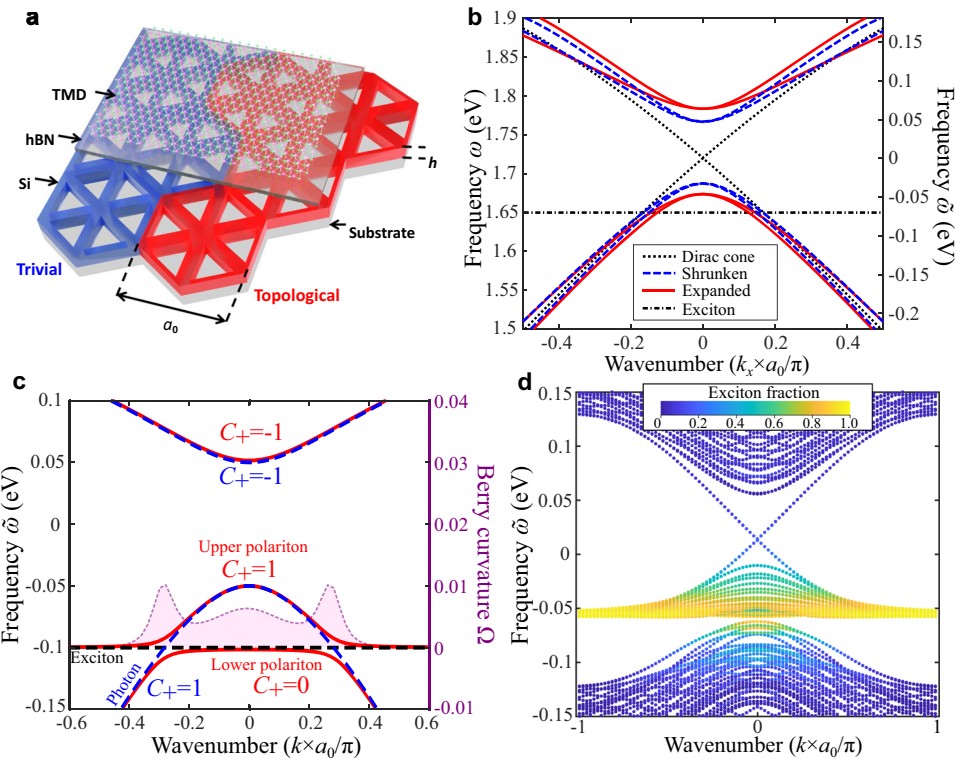

**Fig. 1 Topological polaritonic metasurface integrating transition metal dichalcogenide (TMD) monolayer. a** Schematic image of topological metasurface with hexagonal boron nitride (hBN) spacer and MoSe₂ monolayer on top. Lattice constant is $a_0 = 460$ nm, Si layer thickness is $h = 75$ nm. **b** First-principle calculated photonic band structure for the cases of gapless (black dotted lines), topological (red lines) and trivial (blue dashed lines) metasurfaces. The spectral position of the exciton is shown by the dashed horizontal line (here, at 1.65 eV in absolute value, which corresponds to exciton in MoSe₂ at low temperature). The righthand $y$-axis is in relative energy units (as measured from the Dirac point) **c** Bulk band structures of topological polaritonic system obtained for one (pseudo-)spin from the analytical model for cases without (blue dashed lines) and with exciton-photon coupling $q_d$ (red solid lines). In the uncoupled case, horizontal black dashed line ($\tilde{\omega}_{ex} = -0.1$ eV here to better illustrate the avoided crossing and the Berry curvature distribution near it) shows the spectral position of exciton. Shaded magenta curve shows distribution of the Berry curvature over the upper polariton band with the corresponding (magenta) $y$-axis on the right side. **d** Excitonic fraction (color coded) of the band structure calculated by TBM on a supercell lattice with topological and trivial domains (10 unit cells each) separated by domain walls for the case of optimal (for maximal excitonic fraction of the edge states) crossing scenario near the $\Gamma$ point.

$$
\widehat{\mathcal{H}}_k = B_2|\mathbf{k}|^2\hat{I} + \begin{pmatrix} M - B|\mathbf{k}|^2 & A(-ik_x + k_y) & 0 & 0 \\ A(ik_x + k_y) & -M + B|\mathbf{k}|^2 & 0 & 0 \\ 0 & 0 & M - B|\mathbf{k}|^2 & A(-ik_x - k_y) \\ 0 & 0 & A(ik_x - k_y) & -M + B|\mathbf{k}|^2 \end{pmatrix}. \tag{4}
$$

Here, an energy shift $\omega_0$, equal to the frequency of the Dirac point for the unperturbed lattice, was introduced so that the Dirac point arises at zero energy. In what follows we express frequency in electron-volt units. In Eqs. (1–3) $\hat{a}_{\mathbf{k},l}$ and $\hat{b}_{\mathbf{k},l}$ ($\hat{a}^+_{\mathbf{k},l}$ and $\hat{b}^+_{\mathbf{k},l}$) are the annihilation (creation) operators for photons and excitons, respectively, and the label $\mathbf{k}$ corresponds to the Bloch momentum of photonic modes, $\tilde{\omega}_{ex} = \omega_{ex} - \omega_0$ is the exciton frequency shifted by $\omega_0$, and $g_{(\mathbf{k},l,m,s)}$ describes the coupling between photonic and excitonic degrees of freedom in our system. We note that the subscript $m = \pm 1$ simultaneously describes the orbital momentum of excitons and their valley degree of freedom (K or K') due to the valley polarization in the TMD monolayer. As described in Supplementary Notes 1 and 2, the $4 \times 4$ structure of the photonic Hamiltonian Eq. (4) incorporates both pseudo-spin $s$ and orbital momentum $l$ degrees of freedom of photonic modes, while the mass term $M$ reflects the band inversion.

The exciton-photon coupling and its form are crucial for generating the TPs reported here, as it induces the topological charge transfer from a photonic to a polaritonic band. In our case, this coupling, described by the $2 \times 4$ matrix

$$
g_{(\mathbf{k},l,m,s)} = \begin{pmatrix} q_{p_+} & q_{d_+} & 0 & 0 \\ 0 & 0 & q_{p_-} & q_{d_-} \end{pmatrix}, \tag{5}
$$

ensures that angular momentum is conserved, which is also reflected by the $\delta_{s,m}$ factors in Eqs. (1–3), and $q_{p_+} = q_{p_-}$, $q_{d_+} = q_{d_-}$ (when the TR symmetry is preserved). This ensures that the spin-orbital coupling in the original photonic system is effectively transferred to exciton-polaritons once the hybrid system is in the strong coupling regime.

The form of coupling in Eq. (5) ensures preservation of the two TR partners for excitons and photons, and it also ensures that the coupling takes place only between pseudo-spin-up (-down) photons and spin-up (-down) excitons. Consequently, the presence of spin-orbital coupling in the photonic metasurface leads the indirect coupling between the orbital momentum of photons and the spin of excitons. This indirect spin-orbital coupling represents the main mechanism behind the topological polaritonic $Z_2$ phase[70,71] observed here. Thus, the form of coupling described by Eq. (5) can be shown to yield an effective phase winding in the interaction between topological photons and excitons. This winding leads to the transfer of topological

 **3**

invariant from the bulk photonic bands to the bulk polaritonic bands (see Supplementary Note 2 for details).

As a confirmation, in Fig. 1c we show the band structure calculated for the spin-up domain of the effective Hamiltonian Eq. (1) for the case of the ($m = 1$) exciton crossing lower dipolar photonic ($s = 1$) band. For the case of expanded (topological) photonic lattice, without coupling to excitons, the photonic bands are known to possess a spin-Chern numbers $C_{ph} = +1$ and $C_{ph} = -1$ for the lower and upper bands, respectively. Turning on the exciton-photon coupling leads to a transition to the topological polaritonic phase. Thus, we observe that the strong coupling gives rise to avoided crossing of excitonic and photonic bands, and the formation of lower and upper polaritons. Calculation of the spin-Chern numbers for the upper polaritonic band yields nonzero values identical to those of the crossed photonic bands before coupling (as in Fig. 1c for the spin-up upper polariton). Inspection of the Berry curvature in momentum space confirms that the topological transition arises due to exciton photon coupling since the main contribution comes from the region of avoided crossing (Fig.1c and Supplementary Note 2). Similar calculations for the spin-down upper polaritons yield opposite sign of the Berry curvature and of the spin-Chern number, as expected for the $Z_2$ phase.

Band structure calculated for the supercell with two domains, topological and trivial, obtained with the tight-binding model (TBM) (described in Supplementary Note 1) is shown in Fig. 1d, where the degree of the excitonic component of the modes is encoded in the color of the bands. As expected for this scenario, the flat section of the upper polariton (the remnant of the excitonic flat bands) is close to 100% excitonic, and the excitonic fraction fades away as we move into the parabolic portion of the band (having an increasing photonic fraction of the band). For the special case of the excitonic bands touching the tip of the photonic bands ($\omega_{ph}(\mathbf{k} = 0) = \omega_{ex}$) allows for an exact analytical treatment which shows that both upper and lower polaritonic bands appear to be ~50% excitonic at $\Gamma$ point ($\mathbf{k} = 0$). Remarkably, this scenario also yields topological boundary states with largest excitonic component. Indeed, as can be seen from Fig. 1d, the topological polaritonic boundary modes appear to be highly excitonic in a wide range of wavenumbers and energies below the mid-gap frequency, and even have a significant excitonic fraction in the mid-gap and at higher frequencies.

Below we focus on the experimental observation of these phenomena in two scenarios: when such crossing occurs (i) between the lower (dipolar) photonic band and excitons in MoSe$_2$, and (ii) between the upper (quadrupolar) photonic band and excitons in WSe$_2$.

**Experimental results**. The designs of our topological photonic metasurfaces were optimized to exhibit band crossing of the exciton resonances in MoSe$_2$ with lower photonic band and in WSe$_2$ with upper photonic band near the $\Gamma$-point. The final designs were fabricated by patterning Silicon on Insulator (SOI) substrates with the use of e-beam lithography followed by reactive ion etching. The details of the fabrication techniques used can be found in Methods. The fabricated samples consist of shrunken and expanded regions forming an array of armchair-shaped domain walls[67]. The bulk regions of at least 10 periods were confirmed to be wide enough to eliminate possible coupling between the edge states confined at the domain walls. One of such domain walls separating topological and trivial regions is shown in an SEM image in Fig. 2a.

Optical microscope images of the samples prepared for lower and upper band crossing scenarios are shown in Fig. 2b and c, and correspond respectively to (i) the case of the MoSe$_2$ monolayer on top of the metasurface with subsequent transfer of a 12 nm hBN layer and (ii) the case of a WSe$_2$ monolayer incapsulated by a 10 nm (bottom) and 30 nm (top) hBN layers. The boundaries of both TMD monolayers and hBN flakes on top of the metasurfaces are indicated by color lines in Fig. 2b, c. The leaky character of the photonic and polaritonic bands allows their optical characterization by the back focal (Fourier) plane imaging in our custom-built experimental setup, which enables the extraction of the band diagrams in frequency-momentum space in a range from cryogenic to room temperatures (see Methods for details on the experimental techniques used).

We observe the formation of TPs by comparing the angle-resolved differential reflectivity spectra from different domain walls of our metasurface (Fig. 3). The data from a domain wall without MoSe$_2$ (Fig. 3a) reveal the photonic bandgap of the metasurface (~1.65–1.77 eV, see Supplementary Note 6 for details on experimental data processing) with two gapless modes inside the bulk bandgap, which correspond to the pseudo-spin-up and pseudo-spin-down topological photonic edge states propagating along the domain wall in the opposite directions. In this case, no polaritonic bands appear, yielding purely photonic topological edge states. In contrast, the spectra measured at 7 K from the domain wall covered by MoSe$_2$ (Fig. 3b) demonstrate strong coupling between the lower energy photonic band of the topological metasurface and the exciton, which crosses it close to the band edge (1.65 eV) and gives rise to the transition to the topological polaritonic phase. The experimental spectra reveal the formation of polaritonic bands with Rabi frequency of $\Omega_R = 27.3$ meV (exact value obtained by fitting the PL data as shown further). This is corroborated by the cross-polarized reflectivity data (Fig. 3c) which has greatly enhanced contrast due to suppression of the background Fabry–Perot interference from the oxide layer. These data confirm the formation of TP bulk states as well as topological edge polaritons. As an important indication of the polaritonic nature of the edge states at 7 K in Fig. 3b, c, we notice that the respective bands asymptotically approach the polaritonic bulk band. The dispersions of the edge modes extracted from the measured cross-polarized reflectivity maps fit well with the dispersion calculated via TBM (Fig. 3d). In agreement with theory (Fig. 1b) and the bulk boundary correspondence[71,72], this confirms that the topological invariants, i.e., the spin-Chern numbers, were transferred to the respective upper polaritonic bands from the former photonic band due to the strong coupling. Accordingly, the lack of edge states within the bandgap between upper and lower polariton branches further evidences that the spin-Chern number of the lower polariton band is zero due to this transfer. Notably, the transition from topological photonic to topological polaritonic spin-Hall effects can also be observed by changing the temperature, which influences the spectral position and oscillator strength of the exciton (see Supplementary Fig. 8).

We also note that for some experimental spectra the presence of a small gap between forward and backward edge states is evident at normal incidence (e.g., Supplementary Fig. 6). While the presence of such a gap is known from the original theoretical study[64], here, the variation of the gap width between different samples allows us to conclude that the primary mechanism of its formation is backscattering due to defects in the structure causing the coupling between backward and forward edge modes. At the same time, the narrow width of the gap (which varies across different samples, but generally does not exceed 5 meV) shows that this backscattering is very weak, and the presence of the defects does not affect the observed topological polaritonic phase.

We further study the TPs via angle-resolved photoluminescence (PL) at 7 K. In the experiment, the sample was excited by HeNe continuous-wave laser (1.96 eV). The spectra shown in

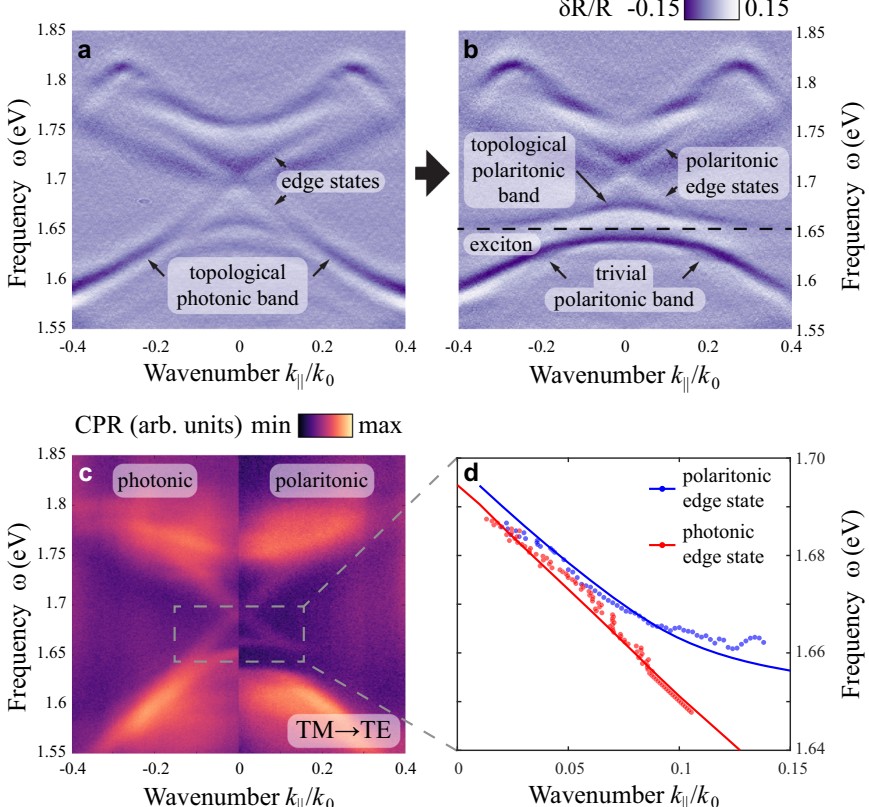

**Fig. 2 Experimental samples of topological polaritonic metasurfaces. a** SEM images of topological photonic metasurface with unit cells of trivial and topological domains indicated by hexagons and the domain wall shown by the armchair shaped black line. Optical microscope images of the two topological polaritonic metasurface samples (black) with TMD monolayers (orange) and hexagonal boron nitride (hBN) spacers (green, crimson). **b** MoSe$_2$ monolayer transferred directly onto the metasurface and covered with a 12 nm hBN flake. **c** WSe2 monolayer incapsulated by 10 nm (bottom) and 30 nm (top) hBN flakes on top of another metasurface.

**Fig. 3 Formation of topological polaritonic bands and edge states with the onset of strong coupling with MoSe$_2$ exciton.** Normalized angle-resolved differential reflectivity measured in the vicinity of two domain walls of a metasurface for transverse magnetic (TM) polarization. The map at the domain wall covered only with hBN (**a**) represents purely photonic scenario, while at the domain wall with MoSe$_2$/hBN heterostructure (**b**), TPs are formed due to onset of strong coupling regime at 7 K. The spectral position of MoSe$_2$ exciton (1.65 eV) is marked with horizontal dashed line. **c** Cross-polarized reflectivity (TM excitation, transverse electric (TE) detection, circularly polarized reflectivity (CPR)) maps in logarithmic scale revealing the modification of the edge state dispersion between photonic (left) and topological polaritonic (right) regimes. **d** Dispersion of the edge modes extracted from cross-polarized reflectivity maps for photonic edge state (red dots) and at 7 K in strong coupling regime (polaritonic edge state, blue dots) compared with the edge state dispersion calculated with tight binding model with and without coupling to the exciton (blue and red lines, respectively).

Fig. 4a, b reveal the emission from both polariton branches as well as from uncoupled neutral exciton at 1.65 eV and charged exciton (trion) at 1.62 eV. We analyze the data by fitting it with the coupled oscillator model using the spectral position and linewidth of uncoupled exciton ($\tilde{\omega}_{ex} = \omega_{ex} + i\gamma_{ex}$, $\gamma_{ex} = 12$ meV) and photonic mode ($\tilde{\omega}_{ph} = \omega_{ph} + i\gamma_{ph}$, $\gamma_{ph} = 10$ meV) that we extract from PL data at large k-vectors and from reflectivity data with no

strong coupling, respectively. The resulting spectral positions of the upper and lower polariton bands are given by the equations[73]

$$\omega_{\pm} = Re\left[\frac{\tilde{\omega}_{ex} + \tilde{\omega}_{ph}}{2} \pm \frac{1}{2}\sqrt{4\kappa^2 + (\tilde{\omega}_{ph} - \tilde{\omega}_{ex})^2}\right], \quad (6)$$

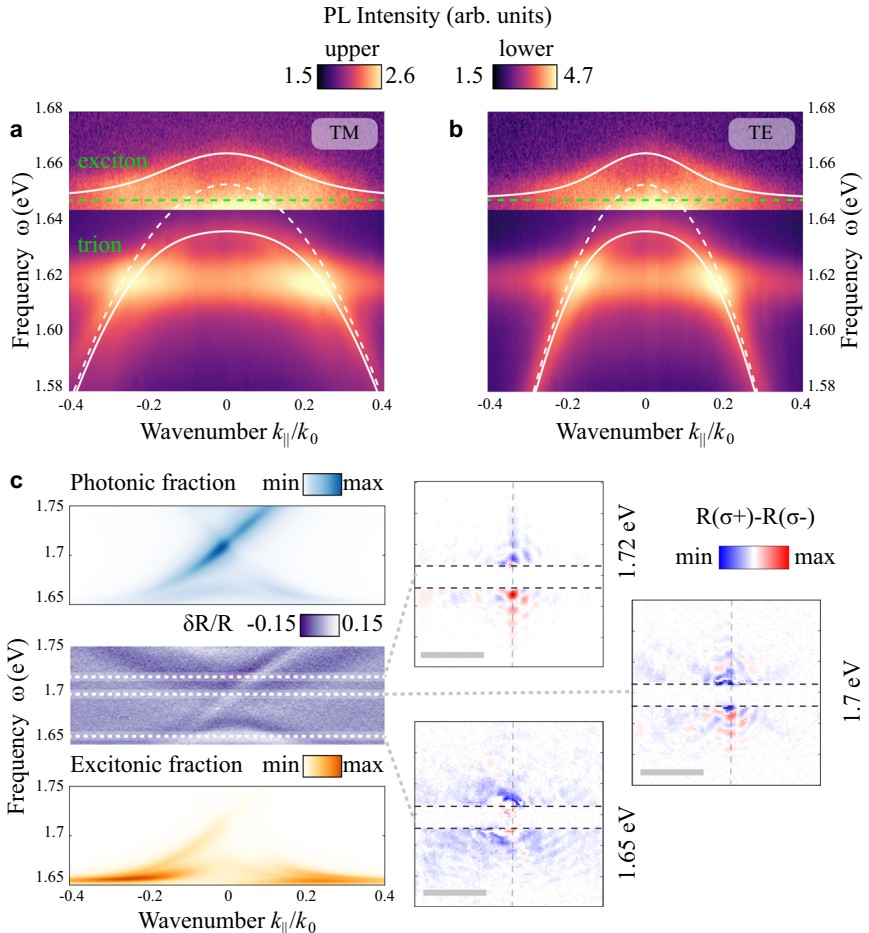

**Fig. 4 Photoluminescence and propagation of MoSe₂ TPs.** Angle-resolved photoluminescence maps for TM-(**a**) and TE-(**b**) polarized collection in logarithmic scale. The sample is excited non-resonantly with CW laser at ~1.96 eV. For better visibility, regions of lower and upper polariton branches are plotted in different color scales. Dispersion of the modes extracted from fitting the PL data with coupled oscillator model are marked with lines: uncoupled photonic mode (dashed white), uncoupled exciton (dashed green) and the resulting polariton branches (solid white). **c** Left: angle-resolved differential reflectivity map for $\sigma^-$ -polarized light compared to the calculation of photonic and excitonic fractions in the modes *via* TBM model. Right: real-space maps demonstrating one-way propagation of polaritonic edge states. The maps show the differential images of the sample excited with focused $\sigma^+$ and $\sigma^-$ -polarized laser pulses with a linewidth of ~10 meV. The calculated exciton fractions for the shown frequencies are 0.05 (1.72 eV), 0.11(1.7 eV), and 1 (1.65 eV, the exciton frequency). Scale bars are each 10 μm. Horizontal black dashed lines enframe the beam stop that blocked the directly reflected light. Vertical dashed lines represent the position of the domain wall.

$$\Omega_R = 2\sqrt{\kappa^2 - \frac{(\gamma_{ph} - \gamma_{ex})^2}{4}}. \qquad (7)$$

From data fitting for both TE and TM polarized PL using Eqs. (6) and (7), we extract a coupling strength $\kappa$ of 13.7 meV, which corresponds to a Rabi splitting $\Omega_R$ of 27.3 meV consistent with the strong coupling criterion.

The most important property of spin-Hall topological systems is the one-way spin-polarized character of their topological boundary states. While it was observed experimentally in photonic structures[74], a similar effect should also emerge in spin-polarized topological polaritonic boundary modes. We demonstrate this experimentally for TPs through circularly polarized reflectivity measurements, where only one of the edge states emerges for circularly polarized excitation of the structure. Figure 4c (left, central panel) shows the one-way edge state in the angle-resolved differential reflectivity map for $\sigma^-$ polarization (see also Supplementary Fig. 7a, b). Using the numerical model of the metasurface supercell with 10 topological and 10 trivial unit cells separated by the domain wall based on TBM/CMT modes, we calculate the excitonic and photonic fractions of the edge mode (shown in Fig. 4c as well). As expected, it has a strong excitonic component, which reaches nearly 100% near the exciton resonance (1.65 eV) and gradually fades away at higher energies where the photonic component starts to dominate. Due to the very low signal-to-noise ratio of PL at the edge modes in this configuration, it was impossible to visualize the propagation of edge topological polaritons for non-resonant excitations. Instead, we corroborate the angle-resolved data with differential real space images of the domain wall of topological polaritonic metasurface excited resonantly by focused circularly polarized laser pulses of the opposite helicity (Fig. 4c, right). As expected, towards the center of the topological gap (1.7–1.72 eV, cf. 1.65 eV), where the contribution of bulk TPs vanishes, the images show clear asymmetry, which indicates one-way propagation of edge TPs along the domain wall. The calculation indicates that at these frequencies the edge modes have considerable excitonic fraction (0.05 and 0.11 at 1.72 and 1.7 eV, respectively).

Finally, we demonstrate the possibility of polarization conservation and selective coupling of valley polarization to the edge

modes. To this aim, we fabricate another topological polaritonic metasurface optimized for strong coupling between the upper photonic band and the exciton in WSe$_2$. The change of material is stipulated by much superior valley depolarization properties of WSe$_2$[74] compared to MoSe$_2$[75]. To characterize the valley polarization conservation in edge TPs, we transfer an hBN-incapsulated WSe$_2$ monolayer on top of the topological metasurface (Fig. 2c) with design optimized for the intersection of WSe$_2$ exciton with the upper photonic band. First, we characterize the sample by measuring the angle-resolved differential reflectivity and PL which confirm the formation of upper and lower TP bands (Fig. 5a, b). As WSe$_2$ emission spectra are dominated by localized excitonic states at low temperature, for the strong coupling characterization we use a temperature of 100 K, for which the polaritonic branches become more pronounced for the PL signal (Fig. 5b). For excitation, we use a 1.96 eV HeNe CW laser. Similarly to MoSe$_2$ sample, we fit the PL data with coupled oscillator model that yields slightly lower coupling strength ($\kappa = 11$ meV) and Rabi splitting ($\Omega_R = 22$ meV) with $\gamma_{ex} = 13$ meV and $\gamma_{ph} = 12$ meV, which can be explained by the presence of an additional hBN layer between the monolayer and metasurface. Next, for the measurements of polarization conservation of TMD emission and its selective coupling to edge states, we employ resonant excitation at the WSe$_2$ exciton frequency (1.74 eV) and decrease the temperature back to 7 K (Fig. 5c, d). Emission from TMD monolayer below the exciton energy (1.68 eV and below, also visible in PL map in Fig. 5b) in such configuration partially

retains valley polarization[76], which provides the opportunity to observe the selective coupling of PL to the edge modes. Figure 5c, d show the maps of angle-resolved differential reflectivity and circular polarization degree (CPD) of emission from edge states of topological polaritonic metasurface with WSe$_2$ measured at 7 K. We define the circular polarization degree as CPD = $(I(\sigma^+) - I(\sigma^-))/(I(\sigma^+) + I(\sigma^-))$, where $I(\sigma^+)$ and $I(\sigma^-)$ are the PL emission intensity recorded for $\sigma^+$ or $\sigma^-$ circularly polarized pump, respectively, without any polarization selective optics in the detection channel. Due to the polarization conservation of WSe$_2$ PL and the spin-momentum locking of the edge modes, the emission is predominantly coupled to one of the counter-propagating edge modes depending on the helicity of the excitation, which leads to the opposite signs of CPD at the edge modes. Figure 5d reveals up to 20% CPD of the emission at the TP edge states, which suggests the possibility of valley transport with topological edge polaritons formed due to the transfer of topological invariant from photonic to polaritonic bulk modes.

## Discussion

To summarize, here we have introduced an approach to engineer $Z_2$ topological polaritonic phase with preserved TR symmetry by strongly coupling a topological photonic system with excitons in 2D materials. The strong polarizability and the presence of two TR partner exciton states in 2D semiconductors, leads to strong coupling and avoided crossing behavior accompanied by the

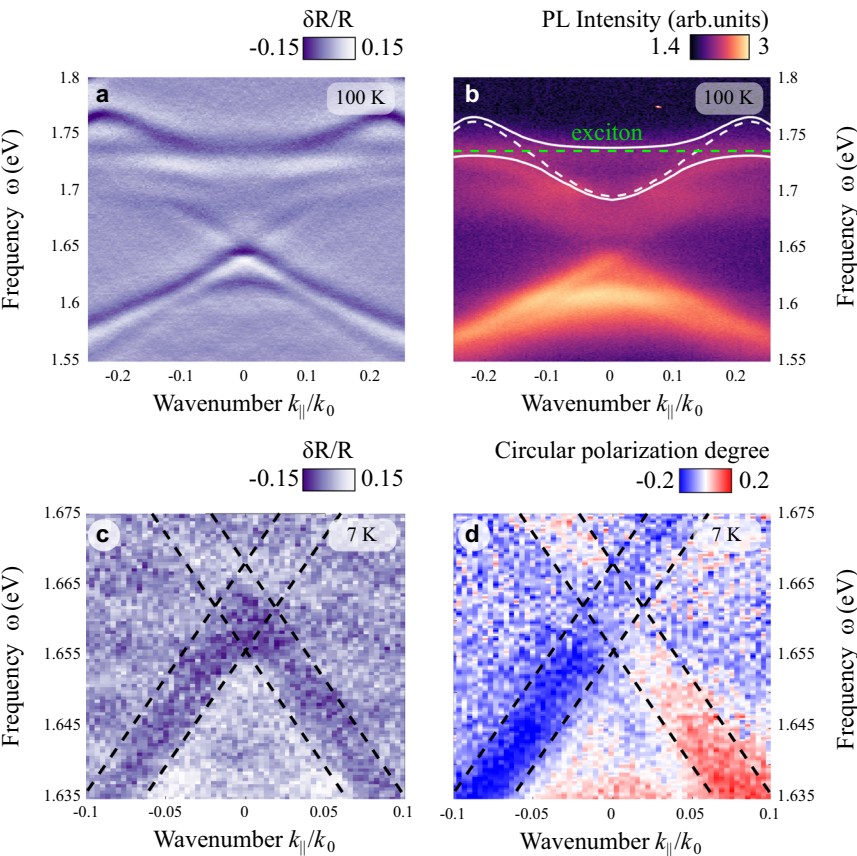

**Fig. 5 Transport of valley polarization by edge TPs. a** Differential reflectivity of the topological metasurface with WSe$_2$ measured at 100 K. **b** Corresponding angle-resolved photoluminescence map for non-resonant excitation at 1.96 eV in logarithmic scale. **c** Angle-resolved differential reflectivity of the metasurface with WSe$_2$ measured at 7 K showing the zoomed-in region with edge states (marked with dashed lines). **d** Degree of circular polarization of emission from WSe$_2$ for resonant excitation at the energy of WSe$_2$ exciton at 7 K (1.74 eV). Edge modes are additionally confined with dashed lines in **c** and **d**. Photonic design parameters are as follows: $a_0 = 488$ nm, $h = 70$ nm.

emergence of effective winding and topological transition to the topological polaritonic spin-Hall phase. TR symmetry and conservation of angular momentum in the system ensures that the excitons with opposite orbital momenta couple with photons of respective pseudo-spins, thus ensuring the formation of two TR partner topological polaritonic bulk bands carrying nonzero spin-Chern numbers. This gives rise to the emergence of spin-polarized one-way edge TPs, which may carry valley-polarized polaritonic component.

Our work demonstrates $Z_2$ 2D topological polaritonic phase which does not require magnetic field and exhibits a large topological bandgap hosting one-way topological polaritonic boundary modes. In our work the topological bandgap amounts to 20–30 meV, which is especially important for potential applications leveraging topological polariton nonlinearities, e.g., for generation of topological solitons which require a broad range of frequencies.

We note that in a recent work the formation of edge exciton polaritons has been reported[77] due to hybridization of $WS_2$ excitons with the topological photonic states of a SiN metasurface. In our work we focus on a different scenario of strong coupling with bulk photonic modes, which gives rise to a transfer of topological invariant to bulk polaritonic states. The resultant $Z_2$ type topological polaritonic phase gives rise to the formation of topological edge polaritons, whose dispersion asymptotically approaches bulk polaritonic bands. Interestingly, in the case of the interaction between photonic edge states and excitons[77], the bulk states remain photonic, and the mechanism of edge polariton formation is different as it gives rise to the gapped character of edge polariton dispersion, with edge polaritons exhibiting one-way spin-polarized propagation. In the scenario considered here, we confirmed that for the case of hybridization of excitons with bulk photons, the edge polaritons emerge from the bulk upper polaritonic bands, in agreement with the bulk-boundary correspondence, which directly evidences the $Z_2$ topological polaritonic phase.

Our work paves the way to engineering topological phases in hybrid photonic-excitonic structures by enriching these systems with additional degrees of freedom inherited from their solid-state component, such as valley degree of freedom in TMDs. Our results thus envision an original platform that can be employed as a resilient topological interface between photonic and electronic components in future valleytronic devices. This platform also has clear advantages over the conventional approach based on semiconductor heterostructures since 2D materials support a broad range of excitations with a variety of internal degrees of freedom and are easy to integrate into topological photonic systems. Thus, this concept can be extended to a wide range of solid-state systems hosting different excitation, including phonons, polarons, magnons and spin-waves, which can be devised to interact with various topological photonic systems, e.g., regular and higher-order topological insulators, yielding topological phases and ways to control matter with light in a robust and a resilient manner. Strong and resilient light-matter interactions in such systems will facilitate enhanced nonlinear effects and novel quantum effects involving half-light and half-mater excitations, which can be of immense value for various classical and emerging quantum applications.

## Methods

**Sample fabrication**. A triangular lattice formed by the hexamers of triangular-shaped holes was fabricated on the Silicon-on-Insulator substrates (75 nm of Si, 2 μm of $SiO_2$) with the use of E-beam lithography (Elionix ELS-G100). First, the substrates were spin-coated with e-beam resist ZEP520A of ~170 nm thickness and then baked for 4 min at 180 °C. Next, gold film 15 nm thick was sputtered on top of resist. E-beam lithography exposure was followed by gold etch and the development process in n-Amyl Acetate cooled to 0 °C for ~35 s. Then, anisotropic plasma

etching of silicon was conducted in The Oxford PlasmaPro System ICP by a recipe based on C4F8/SF6 gases. Triangular shaped holes were etched to the depth of about 75 nm at temperature 5 °C etching with rate about 1.5 nm/s. Finally, the residue of resist was removed by sample immersion into NMP heated to 60 °C.

Monolayers of TMD materials ($MoSe_2$, $WSe_2$) were exfoliated onto a thick PDMS stamp using standard tape technique and transferred to the substrate by the custom-built transfer system. The monolayer was annealed at 350 °C for 2 h to remove the polymer residue from the transfer process. Further, some of the monolayers were encapsulated within hBN layers and annealed again at 350 °C for another 2 h.

**Experimental set up**. Angle-resolved reflectivity measurements were performed in a back focal plane configuration with a slit spectrometer coupled to a liquid-nitrogen-cooled imaging CCD camera (Princeton Instruments SP2500+PyLoN), using white light from a halogen lamp for illumination. The sample was mounted in an ultra-low-vibration closed-cycle helium cryostat (Advanced Research Systems) and maintained at a controllable temperature down to 7 K. To resolve the topological edge modes, we used a slit-type spatial filter in the image plane of the detection channel that allowed to collect signal only from the vicinity of a single domain wall. For polarized reflectivity maps, the polarization selection was performed both in excitation and in collection channels. The maps for aligned polarizers were additionally post-processed to suppress the Fabry-Pérot background originating from the bottom silicon layer of the SOI substrate (see Supplementary Note 6). For PL CPD measurements with $WSe_2$ sample, the linear polarizer was removed from the collection channel to exclude any polarization sensitivity that interferes with CPD extraction.

## Data availability
The data that support the findings of this study are available from the corresponding author upon reasonable request.

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

## Acknowledgements

The work was supported by the National Science Foundation with grants No. DMR-1809915 and NSF QII TAQS OMA-1936351, by the Defense Advanced Research Project Agency Nascent Program, by the Office of Naval Research with grants No. N00014-21-1-2092 and N00014-19-1-2011, and by the Simons Foundation. The work on transferring TMD monolayers and hBN crystals on top of metasurfaces was supported by the Ministry of Science and Higher Education of Russian Federation, goszadanie no. 2019-1246 and megagrant 14.Y26.31.0015. Optical measurements were funded by RFBR, project number 20-32-70185. Optical measurements on WSe₂-based photonic structures were funded by the Russian Science Foundation (project no. 19-72-20120). DNK and MSS acknowledge the support from UK EPSRC grants EP/R04385X/1 and EP/N031776/1. IS acknowledges the support from The Russian Federation President Grant Council, project no. MK-4652.2021.1.2.

## Author contributions

M.L., I.S., F.B., T.I., E.K., S.K., A.V., S.G., M.S., V.M., D.K., A.A., A.S., A.B.K. contributed extensively to the work presented in this paper, preformed theoretical and experimental studies, and prepared the manuscript. M.L. and I.S. contributed equally to this work.

## Competing interests

The authors declare no competing interests.

**Additional information**

