## [Peer Review File · Nature Communications]

REVIEWERS' COMMENTS

Reviewer #1 (Remarks to the Author):

The authors have further improved the manuscript, in particular regarding its comprehensiveness and readability. I appreciate these efforts. I also think, that the newly added discussion in the concluding section of the paper, which puts the current work in relation to ref. 80 is useful for the reader.

Based on these modifications, I recommend publication of the work.

Reviewer #2 (Remarks to the Author):

I have been reviewing the third version of the present manuscript which is now considered for publication in *Nature Communications*.

I appreciate the efforts made by the authors to reply to the different referee's comments, and the positive opinion about their work I had in my previous report still holds.

Nonetheless, I have a comment regarding the new discussion of the paper by Liu et al. [*Science* 370.6516 (2020): 600-604]:

First, I would like to point out, that while Liu et al. mostly focus on the light-matter hybridisation of the interface modes, they performed measurements demonstrating strong coupling with bulk modes in a second sample and it is shown in their supplementary Figure S16.

Second, I rather disagree with the statement: "the mechanism of edge polariton formation reported in Ref. 80 is very different from ours...". I understand that depending on the position of the exciton energy with respect to the photonic bandgap one obtains different polariton bands and that it has to be accounted properly in the effective model and in the determination of the band topological indices. While it is worth to highlight this difference with the paper by Liu et al. (main text), in the end, the origin of the interface state is the same. In my opinion, this fact should be clear for a general reader, and I think the present discussion is a bit confusing.

Finally, I would like to recall the others differences (which I believe are relevant from an experimental point of view) mentioned in my previous report: Liu et al. results have been obtained with a different TMD material and at a different interface type (zigzag) in the photonic structure.

Response Letter to Reviewers' Reports

Reviewer #1

Reviewer #1 General Remark

The authors have further improved the manuscript, in particular regarding its comprehensiveness and readability. I appreciate these efforts. I also think, that the newly added discussion in the concluding section of the paper, which puts the current work in relation to ref. 80 is useful for the reader.

Based on these modifications, i recommend publication of the work.

Authors response to Reviewer #1 General Remark

We would like to thank the Reviewer #1 for their positive feedback and for their comments which were crucial for improving overall quality of the manuscript.

Reviewer #2

Reviewer #2 General Remark

I have been reviewing the third version of the present manuscript which is now considered for publication in Nature Communications.

I appreciate the efforts made by the authors to reply to the different referee's comments, and the positive opinion about their work I had in my previous report still holds.

Authors response to Reviewer #2 General Remark

We thank the Reviewer #2 for their positive feedback to our work, to the revisions made in previous iterations, and for their additional suggestion. We have adjusted the manuscript accordingly as described below.

Reviewer #2 Specific Comment/Suggestion

Nonetheless, I have a comment regarding the new discussion of the paper by Liu et al. [Science 370.6516 (2020): 600-604]:

First, I would like to point out, that while Liu et al. mostly focus on the light-matter hybridisation of the interface modes, they performed measurements demonstrating strong coupling with bulk modes in a second sample and it is shown in their supplementary Figure S16.

Second, I rather disagree with the statement: "the mechanism of edge polariton formation reported in Ref. 80 is very different from ours...". I understand that depending on the position of the exciton energy with respect to the photonic bandgap one obtains different polariton bands and that it has to be accounted properly in the effective model and in the determination of the

band topological indices. While it is worth to highlight this difference with the paper by Liu et al. (main text), in the end, the origin of the interface state is the same. In my opinion, this fact should be clear for a general reader, and I think the present discussion is a bit confusing. Finally, I would like to recall the others differences (which I believe are relevant from an experimental point of view) mentioned in my previous report: Liu et al. results have been obtained with a different TMD material and at a different interface type (zigzag) in the photonic structure.

Authors response to Reviewer #2 Specific Comment/Suggestion

We thank the Reviewer #2 for pointing out to the experimental data in the Supplement of work by Liu et al. To avoid any confusion, we have revised the respective paragraph accordingly not to imply unintentionally that the strong coupling with bulk states was not demonstrated in the respective work. The revised paragraph now explains that in our work we emphasize a specific mechanism of formation of topological polaritons via hybridization with bulk topological photons and the respective bulk boundary correspondence. We also mentioned different materials used in the two works. We thank the Reviewer #2 again for their suggestion.